# A Multi-Port Hardware Energy Meter System for Data Centers and Server Farms Monitoring

**DOI:** 10.3390/s23010119

**Published:** 2022-12-23

**Authors:** Giuseppe Conti, David Jimenez, Alberto del Rio, Sandra Castano-Solis, Javier Serrano, Jesus Fraile-Ardanuy

**Affiliations:** 1GATV Research Group, Signals, Systems and Radiocommunications Department, Universidad Politécnica de Madrid, 28040 Madrid, Spain; 2Information Processing and Telecommunications Center (IP&T Center), Universidad Politécnica de Madrid, 28040 Madrid, Spain; 3Escuela Técnica Superior de Ingeniería y Diseño Industrial (ETSIDI), Universidad Politécnica de Madrid, 28012 Madrid, Spain

**Keywords:** voltage and current sensing, energy meter, multi-port meter, smart meter, hardware measurement platform, advanced metering platform, embedded system

## Abstract

Nowadays the rationalization of electrical energy consumption is a serious concern worldwide. Energy consumption reduction and energy efficiency appear to be the two paths to addressing this target. To achieve this goal, many different techniques are promoted, among them, the integration of (artificial) intelligence in the energy workflow is gaining importance. All these approaches have a common need: data. Data that should be collected and provided in a reliable, accurate, secure, and efficient way. For this purpose, sensing technologies that enable ubiquitous data acquisition and the new communication infrastructure that ensure low latency and high density are the key. This article presents a sensing solution devoted to the precise gathering of energy parameters such as voltage, current, active power, and power factor for server farms and datacenters, computing infrastructures that are growing meaningfully to meet the demand for network applications. The designed system enables disaggregated acquisition of energy data from a large number of devices and characterization of their consumption behavior, both in real time. In this work, the creation of a complete multiport power meter system is detailed. The study reports all the steps needed to create the prototype, from the analysis of electronic components, the selection of sensors, the design of the Printed Circuit Board (PCB), the configuration and calibration of the hardware and embedded system, and the implementation of the software layer. The power meter application is geared toward data centers and server farms and has been tested by connecting it to a laboratory server rack, although its designs can be easily adapted to other scenarios where gathering the energy consumption information was needed. The novelty of the system is based on high scalability built upon two factors. Firstly, the one-on-one approach followed to acquire the data from each power source, even if they belong to the same physical equipment, so the system can correlate extremely well the execution of processes with the energy data. Thus, the potential of data to develop tailored solutions rises. Second, the use of temporal multiplexing to keep the real-time data delivery even for a very high number of sources. All these ensure compatibility with standard IoT networks and applications, as the data markup language is used (enabling database storage and computing system processing) and the interconnection is done by well-known protocols.

## 1. Introduction

In the past decades, the development of information technologies (IT) has allowed the implementation of new applications such as big data analytics, cloud computing, and the Internet of Things, among others [1]. To support these applications, massive computing facilities have been installed around the world, such as data centers and server farms. To accommodate the intensive use of services such as instant messaging, file distribution, or cloud computing, these facilities consume a huge amount of energy to meet the user’s demands. In addition to high electricity consumption, other issues such as greenhouse gas (GHG)-associated emission fares, disturbances in the electric grid, or facility sustainability have become a serious concern for IT companies [2].

To reduce energy consumption and impacts on global warming of data centers and server farms, different approaches have been proposed [3]. A review of power consumption strategies is presented in [4]. Some works propose controlling and monitoring cooling systems [5,6,7,8], because almost 40% of the energy supplied to the server’s facility is used for refrigeration purposes. Other works focus on server design with the aim of improving the performance of processing elements [9,10,11,12,13]. Implementing these actions improves the performance of the specific element, but does not offer a significant reduction in consumption, because they only act on one element of the installation [4].

To significantly reduce energy consumption and other related issues of IT facilities, the implementation of strategies that consider multiple elements of the installation presents better results than the unique element approach [14]. For example, in [15,16] multiobjective problems are proposed to optimize the energy consumption of cloud data centers considering the workload scheme, the energy cost and the integration of renewable energy. Reference [17] presents an online energy management platform that optimizes the Quality of Service (QoS) requirements of distributed data centers by monitoring the cooling system, network congestion and server’s provisioning via Model Predictive Control (MPC). In the same way, in [18] a self-adaptive algorithm based on the weighted sum method (multiobjective functions are combined into one objective function that assigns weighting coefficients to different objectives) is used to reduce instantaneous power by controlling resource allocation and server’s performance.

A key factor to obtain better results in the multi-element approach is to use sensors to measure the different parameters of IT facilities, as is shown in some works founded in the literature. Reference [19] proposes the so-called Energino system (an Arduino platform) composed of sensors and a software tool that is capable of measuring and controlling energy consumption inside a data center. In [20] a real-time monitoring tool is developed using sensors and open source platforms (Arduino, Rasberry Pi, and Gobetwino) to measure and record environmental conditions in the data center. PowerNetS [21] is a power optimization framework developed to minimize the energy consumption of a data center by measuring and controlling the power of servers, cooling devices, and traffic congestion in a data center network. However, the lack of appropriate plug-and-play systems that allow coordinate all measurements and provide reliable measurements makes it difficult to implement these measurement systems.

As can be seen, the energy consumption in IT services is continuously increasing [22]. Energy efficiency, energy savings, and energy consumption minimization are main trends in the road map for IT enterprises and globally to achieve climate neutrality, leading many individuals, governments, and organizations working to find ways to reduce energy demand in a smarter way. For this purpose, new solutions are being developed to improve energy management. Real-time data acquisition, data analytics, and artificial intelligence have become the most used tools to achieve these goals. However, to our best knowledge, there is still a lack of solutions for large IT infrastructures that are typically monitored as a whole. In this work a complete multiport power measuring system is presented, it is able to acquire real-time data up to 16 power inputs (server) and process them to feed a smart system to manage the consumption.

## 2. Related Work

Various systems and methods similar to the main idea of this work have been proposed in previous years. A review of the state-of-the-art approaches to electrical energy metering has been presented in [23]. The authors present a classification of the various methods and functionalities, also introducing an indication of the cost. In [24] the authors provide a great review work, cataloging energy efficiency methods according to a variety of parameters, such as the measurement and verification process, the prediction and recommendation process, and the distinction between deterministic and data-driven models. After classification, the authors conclude that the measure and verification process seems to have a well-defined structure, the prediction and recommendation process could have a structure lack, while data-driven methods permit to plan effective strategies for the energy demand reduction.

The following works introduce different existing metering systems and power metering approaches. A brief description of the advantages and disadvantages of these works is also presented in Table 1.

The authors propose a low-cost energy meter in [25], using an open-source hardware device to read the power measurement from a sensor. The work focuses on high-precision energy and power quality measurement for low-voltage power systems.

Other open source hardware and software platforms are presented in [26] called Elemental. It enables real-time and historical analysis of a building’s performance. This work focuses on inexpensive low-power wireless sensors and controls, with the goal of connecting existing commercial IoT devices with energy monitoring systems to improve the control of information about building activities.

A micro-device dedicated to monitoring, controlling, and managing home energy consumption has been proposed in [27]. The electronic device is designed to be installed into a Schuko socket. The authors describe the system and its communication method, which uses power lines to exchange energy consumption data between the central node and the microdevices that perform the measurements.

The authors of [28] introduce a method called Novelty Detection Power Meter (NodePM). The hardware of the system uses the Zigbee standard to communicate. The main concept of this work is to detect the novelty of energy consumption of electronic equipment monitored in a smart grid. In fact, NodePM is integrated into a remote monitoring platform and uses a Markov chain model and a machine learning algorithm to monitor electric energy consumption and analyze the behavior of electronic equipment. Additionally, the system can send alerts to a smartphone in case an anomaly is identified.

An architecture that brings Artificial Intelligence (AI) into smart homes for the management of electrical energy has been presented in [29]. The authors developed an Arduino-based edge analytics power meter with push notification service. Power consumption is sent to the cloud using the LINE Notify service. The cloud is used as edge analytics to process the data and apply AI models. The energy management systems developed in this architecture is able to extract data from the monitored electrical appliances, and to reduce the energy need using the different AI methods that were trained.

Research has investigated how to implement power quality metrics (PQ) in a low-cost smart metering platform in [30,31]. The authors collect standard requirements for PQ and harmonic measurements and develop a strategy for the integration of PQ metrics on commercial platforms for smart metering. The commercial device adopted in the test is STCOMET from STMicroelectronics [32]. The studies implement the PQ metrics and perform a PQ analysis that also examines the distorted voltage signals. The studies perform a deep analysis, taking into account up to 25 harmonics and choosing their amplitudes according to the CEI EN 50160 limit values for electrical networks.

A smart meter was presented in [33,34]. The authors designed a non-intrusive device that can be easily installed in the panels. Additionally, harvesting techniques have been adopted to power the device. In fact, it is able to exploit the magnetic field inducted around a wire carrying electricity to perform the measurement and to acquire the power required to operate. Communication technologies have been explored in depth by the authors who follow the study using Sigfox in [33] to transmit power consumption data. Successively, the authors follow the research in [34], analyzing other standards such as LoRaW2A6N6, NB-IoT, Wi-Fi, BLE, and comparing the result achieved with Sigfox.

**Table 1 sensors-23-00119-t001:** Comparison of existing state-of-the-art metering systems.

Work	Advantages	Disadvantages
Viciana et al., 2019 [25]	Small; Low-Cost; Open-Source; Graphic dashboard front end; Energy and power quality measurement; real-time monitoring; communicate the measurement to the cloud.	Not suitable for use in high-voltage systems; single channel global measurement; Implementation and use only for specialized users.
Ali et al., 2019 [26]	Open-Source; Real-time; Low-Cost; Fault detection; Distributed wireless sensor nodes; Graphic dashboard front end; Data stored on a database.	Technical expertise required for set-up and use; Not suitable for use in high-voltage systems; Possible security vulnerability to cyber attacks due to the network wireless sensors in buildings.
Morales et al., 2012 [27]	Easy installation inside a Schuko socket; Use of power-line communication which allows it to be easily installed in both new and old buildings.	Possible susceptibility to interference from other devices on the power grid due to the use of power-line; No graphic front end; Require advanced user for installation and use; Only for low voltage systems.
Filho et al., 2014 [28]	Use of probabilistic techniques and machine learning to identify power consumption patterns in electronic equipment monitored by a smart grid; Use a wireless sensors network; Easy and flexible deployment in a variety of settings.	Not suitable for use in high-voltage systems; Technical expertise required for set-up and use; Possible need of a large amount of computing power and storage capacity for the machine learning algorithms; Possible security vulnerability to cyber attacks due to wireless sensors.
Chen et al., 2019 [29]	Real-time monitoring; Advanced artificial intelligence and cloud analytics for device monitoring; LINE messaging app notifications; Multi appliances monitoring.	Complex technological infrastructure and expertise to implement; Possible latency issues due to the distance between on-site IoT devices and the cloud analytics service; Tested only on low voltage systems.
Artale et al., 2018, 2020 [30,31]	Low-cost smart metering platform; Power quality (PQ) metrics implemented based on harmonics analysis; Easy and economic installation without replacing exiting meters.	Lack of graphical front end; Technical expertise required for set-up and use; The use of external instrument transformers may also impact the accuracy of PQ measurements; Single channel; Only for low voltage applications; Remote connectivity or remote monitoring not specified.
Saavedra et al., 2020, 2021 [33,34]	Low-cost; Energy harvest self-powered device; Non-intrusive IoT device for smart metering easy to install; Multi wireless protocol (Sigfox, LoRaWAN, NB-IoT, Wi-Fi, BLE).	No graphical front end; Technical expertise required for set-up and use; Power harvesting could not work if the magnetic field induced is too low; Possible security vulnerability to cyber attacks due to the network multi wireless protocols in the differen scenarios; Data collection and energy monitoring storage not specified.
Negirla et al., 2020 [35]	Use of data slicing in PLC smart grid networks; Large data files transmission throughout the smart grid; Remote firmware upgrades; Communications even in low-availability networks due to data slicing; Reducing the need for human intervention.	The use of a specific transmission rate that is tuned to the noise levels of the power grid; Could reduce the transmission rate in case of high noise; May require a significant amount of data processing and storage; No graphical front end.

A work that analyzes the large data exchange over the power line is presented in [35]. The authors propose a data-slicing model for large data files to allow secure data exchange over the Smart Grid. Commercial hardware such as the STMicroelectronics (ST) Power Line Communication (PLC) Evaluation smart meter board has been adopted to carry out the experiment. The authors achieved a good transmission rate over a low-power electrical grid through the proposed method that allows data interchange. Furthermore, remote firmware has been performed, which is updated over the power line, performing a similar performance compared to a manual firmware update using an optical probe.

This work focuses on the energy measurement of a data center or a server farm. This scenario is composed of a large number of servers that host Internet and cloud services that are always operative (such as YouTube, Facebook, etc.). This contest requires the deployment of a large number of sensors, one for each server, to monitor the energy consumption of the entire center. Furthermore, the power consumption could be continuous for services where the users are demanding constantly (e.g., Netflix), or punctual for these applications where the users could access a resource on demand (e.g., Apache server). To reduce energy consumption and optimize resource use, many smart management techniques have been implemented that use the latest approaches, such as artificial neural networks, fuzzy logic, etc., as depicted in [36].

These strategies, which allow for a smart decision to reduce energy consumption, require a hardware set-up capable of providing physical measurement of the appliances under analysis. In the data center and server farm cases, a set of measurements is needed to monitor a large number of appliances. In fact, the severs could process data at the same time, running different applications. The availability of energy consumption measurement data is important for analyzing and choosing the right strategy to avoid the waste of computational resources and optimize power consumption.

## 3. Hardware Management System Development

### 3.1. Real Time Data Acquisition System

The main objective of this work is to describe the creation of a complete measurement system step by step. The equipment should be able to analyze the energy consumption of a data center or a server farm and make these power measurements available in the following stages. In fact, once the data have been received, it is possible to elaborate and apply the methods described in [36], for example, to reduce energy needs or to create a user consumption pattern to optimize resource utilization, etc.

Figure 1 shows the operative stages of the system to perform a measurement. Considering a rack of servers, once a measurement is requested, the system needs the ability to select the correspondent sensor, perform the measurements required, and introduce them into packets for the transmission of data until the storage stage. In this work, a local server that saves the measurement data is used, even if more technologies could be adopted for this stage (cloud, remote server, etc.). Even if this study focuses on data centers and server farms, it could be adapted to any scenario that needs a system that can monitor a large number of devices. To provide energy measurements to the system, several solutions need to be analyzed.

Figure 2 shows the scenario under analysis. A rack of servers represents the target to measure, a set of sensors perform the measurements, a system acting as interface is responsible to interpret the measurements and send them to an embedded system that coordinates the hardware layer and provides the energy consumption to the eventual following stages.

As a starting point, it is important to determine which energy measurements are required, in this case the AC voltage, the AC current, the active power and the power factor (pf) represent the basic measurement to evaluate the power consumption as the active power is obtained from (Equation 1).
(1)P=U∗I∗cosϕ
where *U* represents the AC voltage, *I* the AC current, and the cosine of the angle between current and voltage is represented by cosϕ.

### 3.2. Measurement System Components

In this section, all the components necessary to develop the system are selected and described. The sensor represents is the first stage close to the servers. Many technologies are available to measure the energy consumption, in this case the less invasive solution is needed, nevertheless, assuring the correct provision of the measures. Thus, selected devices have been proven to provide true rms values. Additionally, it is important that the measurement is performed without generating any kind of interference into the server power system and without modifying the servers power distribution connections. After an analysis carried out based on how to perform these measures and the type of sensors [37], the current transformer sensor represents the solution adopted because it allows measurement to be carried out with respect to the requirements and proper accuracy.

The current transformer is an analog device that can measure the current flowing through a conductor due to the magnetic field generated by the current. In fact, as shown in Figure 3, applying to this case, the server power chord acts as primary and the sensor acts as secondary. Measuring the current induced in the secondary allow to measure the current that is flowing into the server power chord (primary). More details about the current transformers and its working principle are available in [37,38].

Thank to this type of sensors is possible to measure the server energy consumption properly without induce any interference into the power supply network. Two current transformers are selected for their precision and cost:YHDC SCT-013-010 on the left side of Figure 4, is a 10 A sensor that provides a voltage output in the range of 0–1 V, which allows to measure server power consumption up to around 2000 W. For more information, please refer to [39];Talema AZ-0500 on the right side of Figure 4, is a 25 A sensor that provides a voltage output in a range of 0–2.5 V, allowing the measurement of server energy consumption up to around 5000 W [40];

This work permits to expand the knowledge of energy consumption server by server, also taking into account the services provided and the applications that run on them. The sensors previously described are chosen not only for their technical specifications (range, precision, etc.), but also for their cost. In fact, considering the economic impact could reduce costs because a large number of sensors are needed to monitor a data center. On the other hand, it is important to choose sensors that maintain a low measurement error.

The next stage is the measurement interpretation stage. This is the layer that acts as an interface that receives the sensors output voltage and extracts the energy measurement.

The hardware needed should be compatible with the sensor’s electrical parameters and perform the required measurements (voltage, current, power, and power factor). Many suitable chips are available on the market and three are chosen as possible candidates: Atmel 90E26 [41], ST Microelectronics STPM32 [42], Cirrus Logic CS5490 [43]. These chips have a similar structure and functionalities to process the signal: the analog input that came from sensors enters into an analog-to-digital converter, and a filtering stage is following that prepares the signal for the DSP stage that performs the measurement. Once the DSP stage performed the measurement, it is available through the SPI or UART interface. More details are available in the data sheets referenced above.

The three candidates have been tested, and Atmel 90E26 was finally chosen due to its value for money, type of technology, high measurement precision and flexibility with respect to communication protocols. Furthermore, the two types of sensors fit perfectly with the Atmel 90E26 in terms of electrical connections and precision according to the system that will be measured.

A test circuit with the Atmel 90E26 has been made. The chip needs to be connected with the sensors to measure the current and with the electric network to sense the operation voltage. Two options are considered for connecting to the electrical network to perform voltage measurements:A resistive voltage divider that consist in a series of resistances that allow reducing the lower voltage required to perform the measurement;An AC transformer that reduces the network voltage to a lower voltage needed;

A calibration test has been performed connecting the Atmel 90E26 circuit to a variable resistive load using both options. The test is carried out by varying the load and, consequently, the power consumption. The measurements presented in this work are performed using the YHDC SCT-013-010 sensor connected to the Atmel 90E26 test circuit and, as a reference, the Yokogawa Mini Clamp-on Tester CL120 [44] for current and the Fluke 115 Digital Multimeter [45] for voltage.

The table in Figure 5 shows the measurement performed with the resistive voltage divider connected directly to the power line, while the table in Figure 6 shows the measurement using the AC transformer. From the comparison, it is possible to deduce that both measurements are a valid option in terms of precision. It is important to note that the first solution in Figure 5 represents a serious threat to the entire system. For this reason, the solution adopter is to introduce the AC transformer in Figure 6 because it allows to decouple and isolate the PCB from the AC network, maintaining the system safe even if it introduces a small error.

The last component to define is the embedded system. The requirement in this case is the communication with the sensors interface to read measurements and the possibility to send them over a desired network (internet, cloud, etc.). The boards should have a SPI or UART interface and a network connection such as Wi-Fi or Ethernet. On the one hand, microcontrollers such as Arduino or STM32 boards could also be a good alternatives, on the other hand, a board with a CPU could represent a better choice because the possibility of running an operating system could give better integration opportunities. A device that contains all the interfaces needed is the Raspberry Pi 3. In fact, this single-board computer has the General Purpose Input/Output (GPIO) interface that makes available the SPI and UART protocols, and it also has WiFi and Ethernet interfaces. More information about Raspberry Pi 3 is available on [46]. This represents a good choice because it does not need any additional components, in terms of price and technical specification, and it allows to install a large set of software and libraries resources on its Linux-based operating system.

### 3.3. PCB Design and Hardware Calibration

After the selection of components is carried out, the next stage consists of the design of a PCB board that allows the Atmel 90E26 to carry out the measurements of the different sensors and at the same time connect to the outside world to send the measurements. As many sensors need to be measured, a possible solution is provided that introduces a multiplexer that can connect each sensor to the Atmel 90E26. The CD4097B multiplexer from Texas Instruments represents a good option due to its compatibility with the voltage channel range, its low resistance channel, which does not introduce electrical issues during connection with the Atmel 90E26, and the ability to connect up to eight sensors.

A possible measurement strategy is also represented by the use of two ATM90E26 to connect up to 16 sensors. Moreover, having two chips could improve the measurement speed because when one chip is performing the measurement, the other chip is selecting the connection to the sensor.

This operation mode could be coordinated using the SPI [47] protocol for communication between the Raspberry Pi3 and the PCB, because it is a full duplex protocol that allows the connection of more devices at the same time. In fact, thanks to the SPI protocol, the Raspberry Pi3, which acts as master, could communicate with the two ATM90E26 that act as slaves, enabling the channel to receive the data, selecting from which chip receives the measurement.

Figure 7 shows the simplified PCB connection schematic. In fact, two ATM90E26 are connected with the Multiplexers (CD4097B), that allow to switch among sixteen sensors. All integrated circuits are routed to an SPI connector that will be linked to the Raspberry Pi 3 GPIO port. The embedded system is responsible for choosing the sensor, sending to the multiplexer the command to select the port, and enabling the ATM90E26 correspondent to read the measurement from the sensor and replying by sending the value back.

Once the requirements have been defined and the necessary tests with physical components have been carried out, the circuit schematic that will be used in the final design is drawn. PCB drawing is performed using a specific CAD program called KICAD [48]. Part of the design and assembly processes are depicted in Figure 8.

The next stage is the test and calibration of the PCB necessary for its verification and for the ability to move on to the final phase of connection with the other components of the prototype.

### 3.4. Drivers and Software Design

The Raspberry Pi 3 coordinates the measurement operations. To perform this task, it needs a driver that permits the use of SPI protocol to handle all the commands necessary to the PCB. The C++ programming language has been selected to code the operations into the embedded system. The connection to the PCB is carried out using the GPIO port of the Raspberry Pi 3, where the Broadcom BCM 2835 chip is responsible for managing the I/O interface control functions. A library that manages BCM 2835 and allows access to GPIO pins is available in [49]. Some useful functions for reading and writing over digital I/O are provided, moreover it permits to use SPI and more protocols. The driver is based on the BCM2835 library, which handles the SPI protocol and sends the commands to the PCB. The functionalities implemented in the driver are as follows:Hardware selection, where the chip and sensor channel that perform the measurement are enabled. This is a unidirectional communication case, where the embedded system (master) sends the command to the PCB (slave) that just sets the device as required;Data interchange, which represents the communication between the master (Raspberry Pi3) and the two slaves (ATM90E26 registers). This is a bidirectional communication case where the embedded system performs a complete SPI read/write operation, sending a command to the ATM90E26 registers, which responds with a status or a measurement data. An example is represented by the measurements (AC Voltage, AC Current, Power, and Power Factor) operations, where the Raspberry Pi3 sends the respective command to the ATM90E26 register, which responds sending the measurement back.

During the calibration described in Section 3.3, the main goal is represented by choosing between the resistive voltage divider and the AC transformer. A similar test has been performed during the driver development stage. In this case, the test was performed to adjust the gain parameters according to the two sensors (YHDC SCT-013-010, Talema AZ-0500). The set of power consumption measurements was performed by comparing the ATM90E26 with the Yokogawa CL120 for the current and the Fluke 115 Digital Multimeter for the voltage, adjusting the gain parameters, and reducing the errors until the measurement is close to the reference.

Table 2 and Table 3 show the measures before and after calibration. The parameters considered are the voltage, the current, and the power measured, respectively, with the reference equipment and the ATM90E26 board. To evaluate the precision of the ATM90E26 board with respect to the reference, voltage and current differences are calculated. Additionally, the power measurement error is calculated as this measure is closest to the target that the system has been designed to acquire. This study takes into account normal application servers that have a power consumption range of about 500 W or 1000 W, which are measured with a 10 A YHDC SCT-013-010 sensor. According to the voltage parameter, as depicted in Section 3.3, it is important to remind that in this work the AC transformer has been selected, preferring the PCB decouple from the electrical network over a less accurate measurement. Having an AC transformer reduces the voltage measurement sensibility, for this reason, the gain parameter calibration has better performance over current parameter.

Analyzing Table 2, it is possible to note that ATM90E26 even if it follows the reference, the measures fluctuate and introduce errors that reduce accuracy. In fact, the error is around 5%, while this value increases as power consumption decreases. This represents an adjustment error because this sensor, according to the datasheet [39] is linear and its precision is ±1%. For that reason, as the sensor is working in its range zone, the greatest error depends only on the parameter calibration of the ATM90E26 PCB.

Table 3 shows the measurement after calibration. The values are closer to the reference, and the error has been reduced. It increases more as the power consumption increases. The error is around 2%, which represents good accuracy, while the last measure has the highest error of 3%. In this case, even if the sensor could reach 35 A, remembering that the YHDC SCT-013-010 is a 10 A sensor, a current of 13.3 A could be considered out of range.

Once the PCB has been tested and calibrated, the prototype could be assembled. As the system has been designed for a server farm or data center application, it is important to take into account connections. A standard used for servers up to 10A is represented by connectors IEC 60320 C-13 [50] and IEC 60320 C-14 [51]. The idea is to introduce the original server power cord into the C-14 prototype input, where an internal cable extension that passes through the sensor, connects to the C-13 prototype output. An other external power cord connects the prototype to the server power input. This method allows to connect a server rack to the prototype without modifying or interfering with the power network. The other port needed is the prototype power input. For this purpose, an IEC 60320 C-8 [52] connector has been introduced. This port has been chosen because the prototype does not need a ground connection and because a separate port is needed to not interfere with the appliances measures, a different connector avoids connection errors. For communication purposes, a female RJ45 Ethernet plug [53] has been introduced that allows the creation of a network connection to the embedded system. Figure 9 shows the assembled prototype described as follows:Top right shows the rear side of the prototype with the C-13 and C-14 connectors couples that represent the 16 channels of the meter, the Ethernet connector, and the C-8 connector that provides power to the internal circuits.Bottom right depicts the front side of the prototype with the power button.A zone of left side internal view, represents all connections between the C-13 and C-14 connectors where the sensors are installed (YHDC SCT-013-010).B zone of left side internal view is the 5 V power supply adapter that provides gain to all internal circuits.C zone of left side internal view, is the AC transformer that allows the PCB to sense the operating voltage of the network.D zone of left side internal view, represents the PCB designed with the two ATM90E26 chips.E zone of left side internal view, is the Raspberry Pi 3 embedded system.

### 3.5. System Connection and Verification

The prototype is now ready to be used and tested. The physical connection is easy to perform, as shown in Figure 10. The prototype has been installed on the rack and placed in an accessible place. In this case, it has been placed on the top of the servers, as shown in the left part of Figure 10, but it is possible to install the prototype in other positions. The electrical connection is shown in the right part of Figure 10, where four rows and eight columns of the C-13 and C-14 connectors can be observed. The lower two rows represent channels one to eight, while the higher two rows represent channels nine to sixteen (one column has four connectors that represent two channels). As explained in the previous section, the C-14 connectors are the channel input, while the C-13 connectors are the channel output. In this case also the cable color helps to understand the connections. In fact, the black cables are the connections of the prototype input channels to the electrical network, while the blue cables are the prototype output channels that power the servers.

One of the key points in all data monitoring is being able to correctly send or host them. Once the system is activated, its operation fluctuates between the different sensors to analyze current consumption. However, the system does not store the information, rather it transmits the data to a central server. Data are sent via the REST API to a server hosted on the same network, where the data are stored after each request. Each source takes around 2 s of analysis and response in sending data. The complete process of measurement and sending of information from all sources is performed in about 15 s. If necessary, the speed of the entire process could be increased.

### 3.6. Measurement System Validation

The proposed architecture is related to the definition and implementation of sensorization in hardware systems and the collection of information through control plane monitoring systems, which will therefore allow the acquisition of numerous data on the state of energy management of the architecture. The functional architecture of the system showed in Figure 11 consists of four main blocks:Data collection: responsible for obtaining the generated data that will be taken into account in the system. It includes data in the three possible formats (structured, unstructured, and semistructured), and must contain those generated directly by M2M (machine-to-machine) interactions, etc.Storage layer: it is in charge of optimally saving the collected data, following the central server structure proposed.Processing and analysis module: it is in charge of providing the necessary tools to explore the stored data and obtain from them the value that was sought.Visualization: it is responsible for displaying the data graphically to ensure their understanding.

To validate that the system is capable of detecting energy variations, the use of Phoronix suite [54] is proposed. Phoronix is a software suite that provides a set of benchmarking tools to evaluate server performance, imposing stress on the system at the desired level. The tests could be automated and it is possible to choose them from a repository of 400 different test scripts. The suite is capable of pushing specific components of the PC to the limit, such as the system itself, the processor, or disk writing. An hybrid combination of tests to check the variation of energy is proposed as follows:AsmFish: Test to simulate constant intensive resource applications, which require high energy consumption.Radiance: Alternate moments of high-performance and low-performance processes. This test allows to simulate applications that could variate their processing intensity.Sysbench: Multi-threaded benchmark to stress the system for punctual requests that can reach the maximum utilization of resources.

Thanks to the Phoronix suite, it is possible to provide a complete stimuli to the servers that allows to simulate the power need close to real applications. This permits to verify the prototype working condition channel by channel because it is possible to launch the same or a different test in each server, producing an energy consumption. The prototype global energy measurement is further validated using a Circutor Line-CVM-D32 [55] power analyzer connected to the input of the power line.

## 4. Results

The architecture of the system ensures precise measurements of each power source. This information is verified in real time and transmitted to data storage servers. Table 4 shows a summary of the most significant measurements for each test. The benchmarks trend for Asmfish vary between no load and maximum CPU usage available, Radiance vary between 20% and 80% of CPU resources, while Sysbench generate peaks of medium and maximum CPU usage. The minimum values of Table 4 correspond to the load-free operating system process because the benchmarks do not use the system resources. The medium values represent the use of more CPU threads, while the maximum values correspond to different high-power need situations (constant maximum available for AsmFish, around 80% for Radiance and a peak of 100% for Sysbench).

Figure 12 shows a set of graphs obtained running a complete Phoronix test. Specifically, the top left Active Power figure summarizes the CPU power consumption during a complete battery of tests that fluctuates depending on the charge CPU peak of the different tests. The remaining figures introduce a graphic for each benchmark showing the corresponding CPU usage details.

Figure 13 presents the voltage measurement comparison between the reference and the ATM90E26 PCB. It is possible to observe that even if the AC transformer introduces a low error percentage, the ATM90E26 measures follow the reference.

Figure 14 presents the current measurement comparison for a single channel of the system. The YHDC SCT-013-010 sensor has linear characteristics in its range (0 to 10 A), while the curve reduces its trend when the current exceeds 10 A to a maximum of 35 A, as reported in the data sheet [39]. As is possible to observe in Figure 14, the ATM90E26 curve shows a trend similar to the characteristics of the YHDC SCT-013-010 sensor. In fact, it is very close to the reference curve. Once the current measurement is close to 10 A, the ATM90E26 curve reduces its trend compared to the reference, showing the characteristic of the YHDC SCT-013-010 sensor, as expected.

Figure 15 represents the power measurement comparison for a single channel of the prototype. The ATM90E26 curve should be a mixture of current and voltage characteristic, but in this case, as the current excursion is greater than the voltage excursion, the curve is similar to Figure 14, where ATM90E26 follows the reference when the current measure is in the range and reduces its trend outside the range.

As reported in Section 3.4, the servers to measure arrive up to 500 W and 1000 W. Taking into account the 230 V line network, where the design of the system and calibration have been performed, it corresponds to a current of around 2 A and 5 A. In the server power consumption working zone, according to Table 3 and Figure 14 and Figure 15, the system is performing the measurement in range, with an error lower then 2% in for the 500 W servers and up to 2% for the 1000 W servers. Moreover the measurement has been globally verified with the Circutor Line-CVM-430 D32 power analyzer. In fact, the measures correspond to the sum of the prototype channels measurements.

## 5. Conclusions

Energy efficiency is the key to achieving the climate goals that the world is aiming for. Real-time data through sensing and intelligent systems have proven to be a great combination for improving the consumption and energy behavior of IT and communication infrastructures. In this work, a complete system capable of real-time monitoring of up to 16 power supplies has been presented.

The system has been designed to acquire the voltage, current, active power, and power factor from the power sources of data servers or workstations and provide them to an intelligent system that enables data-driven decisions. By means of a sensor chip and ad hoc electronic developments the system can acquire the aforementioned electric measures, in a range that could be set up between a minimum time of 500 ms up to the desired time, taking advantages of multiplexing.

The system has been calibrated to achieve its maximum precision in the normal operating range of the power supplies, achieving error rates below 2% percent. The final validation of the system has been carried out on a rack with nine servers.

Servers have been forced by the use of the Phoronix benchmark suite to have various power consumption behaviors in parallel. The system has been able to provide the measurements of each of the servers accurately, as the values provided are very similar to those provided by a professional network analyzer.

The designed system is completely external to the equipment to be measured, which allows it to be used to perform measurements in different environments (considering ideally that the maximum performance occurs when the calibration has been performed according to the power of the source).

## Figures and Tables

**Figure 1 sensors-23-00119-f001:**

System operation workflow.

**Figure 2 sensors-23-00119-f002:**
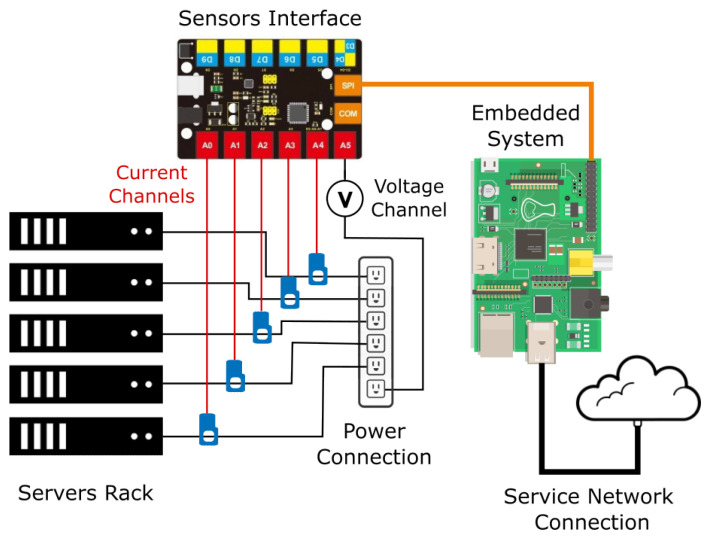
Architecture diagram of the measurement system.

**Figure 3 sensors-23-00119-f003:**
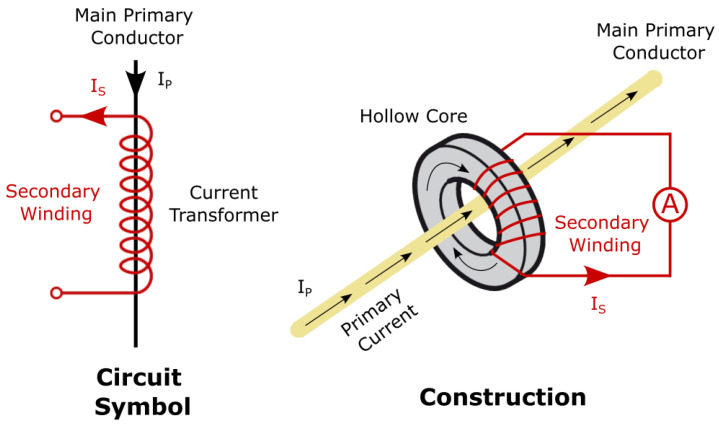
Current transformer working principle.

**Figure 4 sensors-23-00119-f004:**
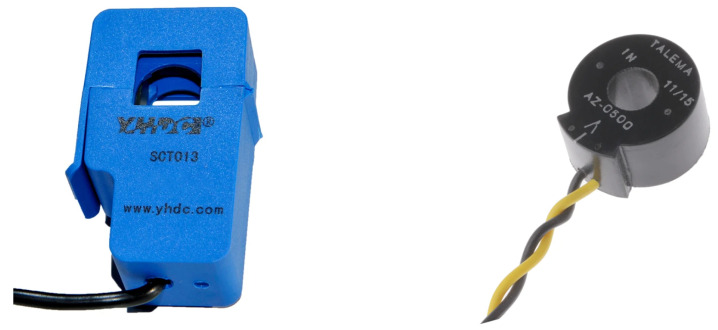
Sensors used in this work: (**Left side**) YHDC SCT-013-010. (**Right side**) Talema AZ-0500.

**Figure 5 sensors-23-00119-f005:**
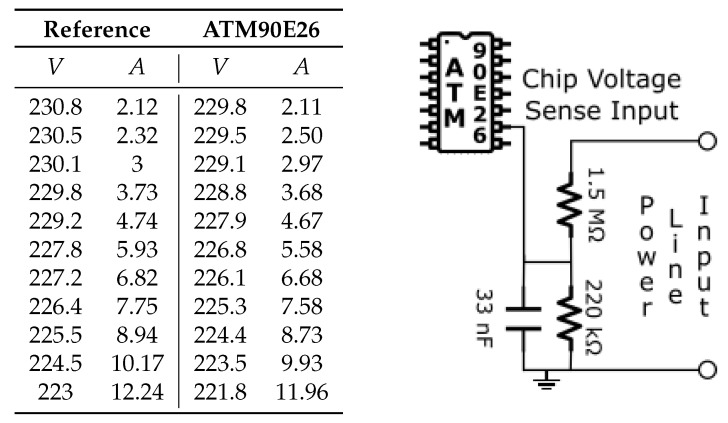
Resistive voltage divider test. Error 2.28%.

**Figure 6 sensors-23-00119-f006:**
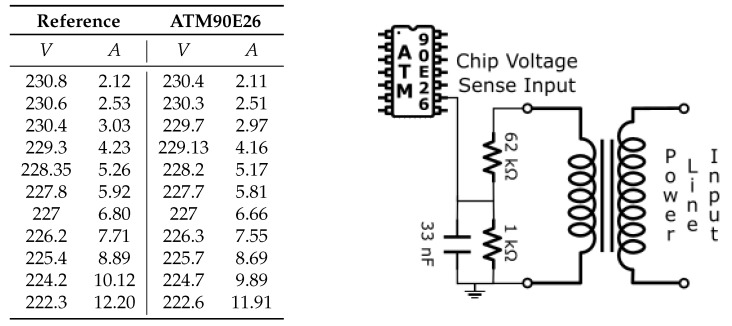
AC transformer and resistive voltage divider test. Error 2.4%.

**Figure 7 sensors-23-00119-f007:**
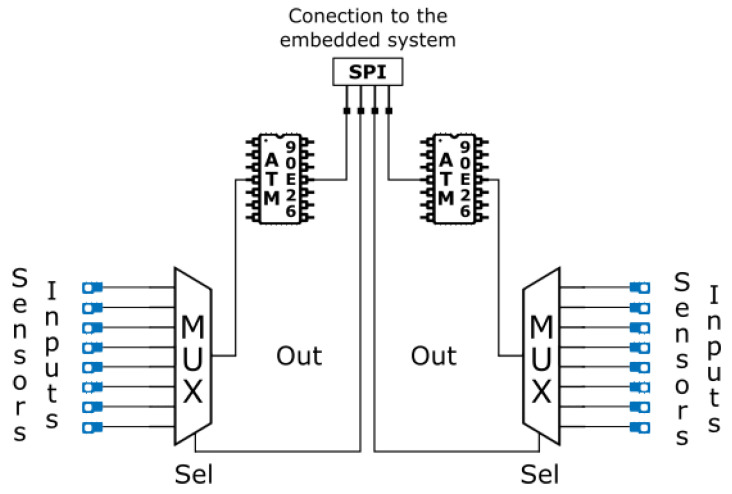
PCB Sensors Interface Diagram.

**Figure 8 sensors-23-00119-f008:**
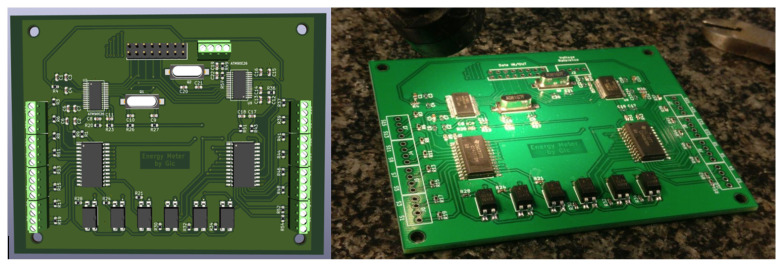
Final design of the PCB. (**Left side**) PCB rendering. (**Right side**) Real prototype.

**Figure 9 sensors-23-00119-f009:**
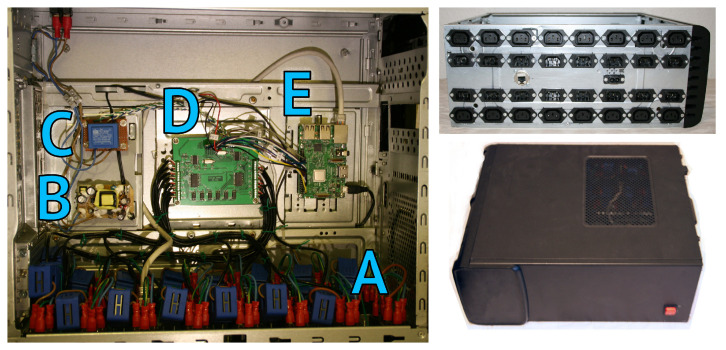
Prototype. Left side: internal view (A sensors zone, B power supply, C AC transformer, D designed PCB, E Raspberry Pi 3), top right: rear view, bottom right: front view.

**Figure 10 sensors-23-00119-f010:**
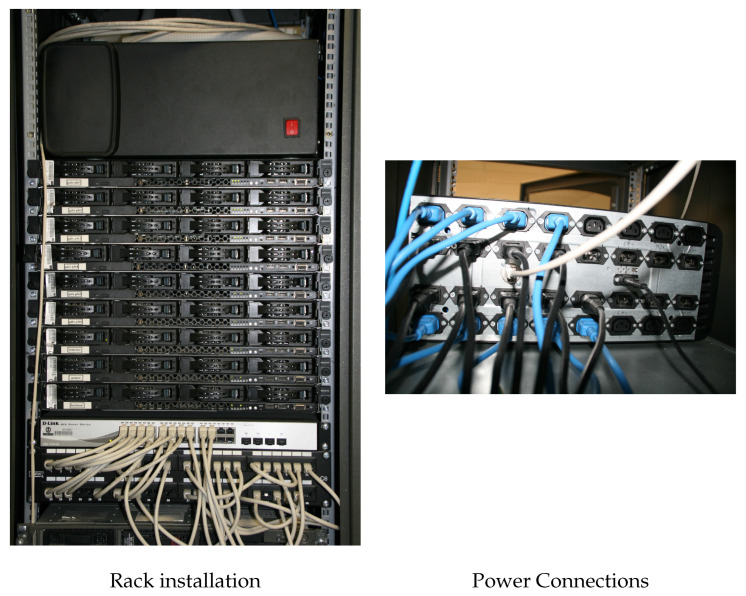
Prototype connection with the servers for energy usage measurement.

**Figure 11 sensors-23-00119-f011:**
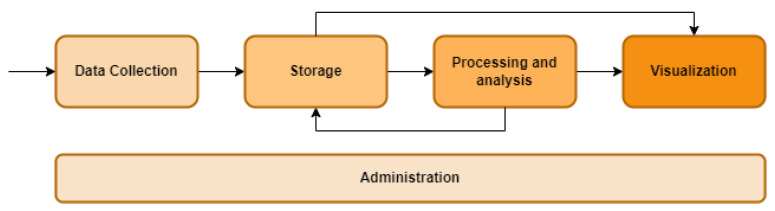
Functional modules of the system for energy consumption.

**Figure 12 sensors-23-00119-f012:**
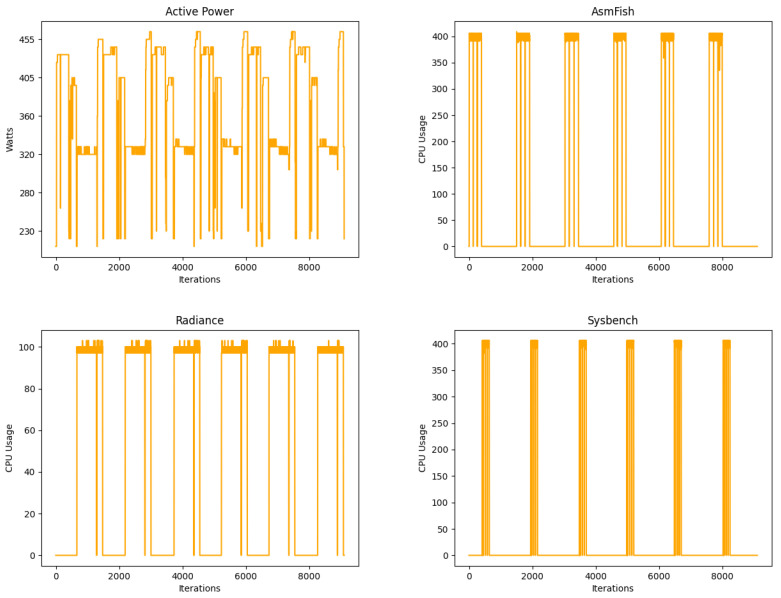
Energy characteristic measurement distribution: (**Top left**) Active Power distribution. (**Top right**) AsmFish test. (**Bottom left**) Radiance test. (**Bottom right**) Sysbench test.

**Figure 13 sensors-23-00119-f013:**
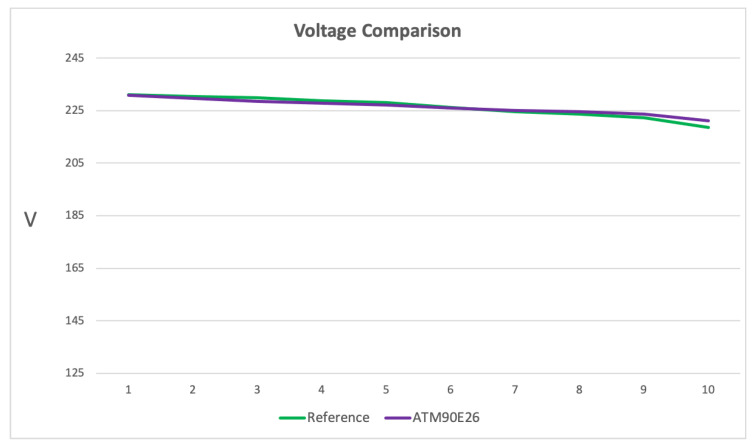
Voltage measurement comparison: Measurements on the vertical axis in Volt, while the horizontal axis expresses the number of measures performed.

**Figure 14 sensors-23-00119-f014:**
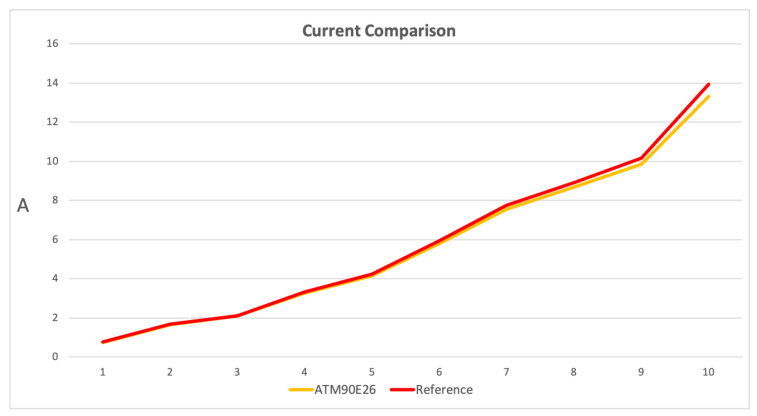
Comparison of current measurements: Measurements are on the vertical axis in Ampere, while the horizontal axis expresses the number of measures performed.

**Figure 15 sensors-23-00119-f015:**
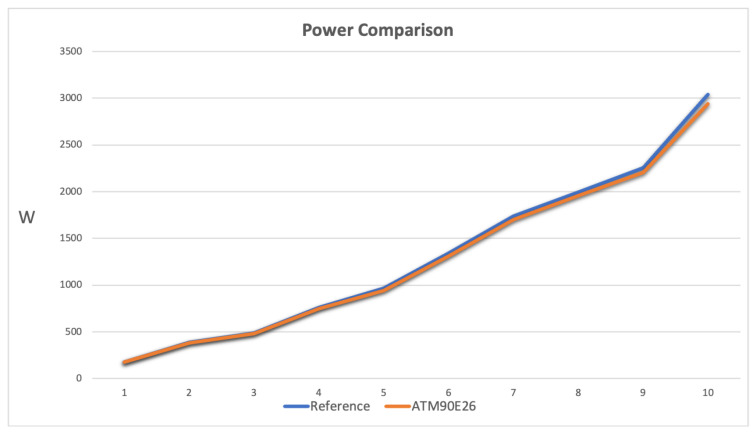
Comparison of power measurements: Measurements are on the vertical axis in Watts, whereas the horizontal axis expresses the number of measures performed.

**Table 2 sensors-23-00119-t002:** Measurement before calibration performed with the YHDC SCT-013-010 sensor.

Reference			ATM90E26			Evaluation		
Voltage (V)	Current (A)	Power (W)	Voltage (V)	Current (A)	Power (W)	Voltage Difference	Current Difference	Error (Power)
231.5	0.46	106.49	229.47	0.51	117.03	2.03	−0.05	9%
230.9	1.08	249.37	229	1.16	265.64	1.9	−0.08	6%
230.8	1.31	302.35	229.1	1.41	323.03	1.7	−0.1	6%
230.2	2.04	469.61	228.68	2.17	496.23	1.52	−0.13	5%
230.9	2.31	533.38	229.1	2.45	561.29	1.8	−0.14	5%
229.4	2.97	681.32	228.7	3.14	718.118	0.7	−0.17	5%
228.9	3.39	775.97	228.37	3.57	815.28	0.53	−0.18	5%
227	4.63	1108.41	227.6	4.87	1051.01	−0.6	−0.24	5%
226.3	5.62	1271.81	226.5	5.88	1331.82	0.2	−0.26	5%

**Table 3 sensors-23-00119-t003:** Measurement after calibration performed with the YHDC SCT-013-010 sensor.

Reference			ATM90E26			Evaluation		
Voltage (V)	Current (A)	Power (W)	Voltage (V)	Current (A)	Power (W)	Voltage Difference	Current Difference	Error (Power)
231.2	0.77	178.02	230.8	0.76	175.408	0.4	0.01	1%
230.5	1.68	387.24	229.7	1.67	383.60	0.8	0.01	1%
230	2.12	487.6	228.4	2.11	481.924	1.6	0.01	1%
228.8	3.33	761.904	227.8	3.28	747.184	1	0.05	2%
228	4.23	964.44	227.26	4.15	943.13	0.74	0.08	2%
226.3	5.93	1341.96	225.97	5.8	1310.63	0.33	0.13	2%
224.6	7.74	1738.40	225.01	7.55	1698.82	−0.41	0.19	2%
223.6	8.91	1953.32	224.52	8.7	1992.27	−0.92	0.21	2%
222.2	10.15	2255.33	223.59	9.86	2204.59	−1.39	0.29	2%
218.5	13.92	3041.52	221.12	13.3	2940.90	−2.62	0.62	3%

**Table 4 sensors-23-00119-t004:** Server rack measurements using Phoronix benchmarks as charge.

	AsmFish				Radiance				Sysbench			
CPU Usage	Voltage (V)	Current (A)	Power (W)	P.F.	Voltage (V)	Current (A)	Power (W)	P.F.	Voltage (V)	Current (A)	Power (W)	P.F.
min	229.22	1.04	202.63	0.85	229.22	1.07	208.47	0.85	229.23	1.06	206.53	0.85
med	-	-	-	-	229.34	1.44	274.11	0.83	229.36	1.87	355.98	0.83
max	229.53	2.6	483.39	0.81	229.39	2.14	402.53	0.82	229.48	2.39	449.73	0.82

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
