# Peer review of "A Multi-Port Hardware Energy Meter System for Data Centers and Server Farms Monitoring"

_sensors, 2022, doi:10.3390/s23010119_

Round 1

Reviewer 1 Report

The paper presents a system capable of real-time monitoring of power supplies. The authors describe a system designed to acquire the voltage, current, active power, and power factor from the power sources of data servers or workstations and provide these data to an intelligent system thus enabling data-driven decisions.

The presentation in the paper is done well, with the research questions and contribution clearly stated. Part of the paper where there is room for improvement is describing the novelty of the research.

The paper is generally well structured and contains appropriate references. The overall level of English is very good. 

Therefore, I would suggest to the authors to discuss the novelty of their research in more detail, although I can even now recommend to accept this paper in its current form.

Reviewer 2 Report

In the presented article, the authors present a measurement system dedicated to broadly understood data processing systems. The work presents the needs of using such systems quite well, however, according to the reviewer, the authors made a mistake by deciding to set cost requirements, i.e. their system is a low-budget system. In the assumptions of the system, it was supposed to perform a measurement (metrological) function and meet the requirements for energy measurement systems, however, the adopted solutions indicate that it can at most perform the function of an indicator. It does not take into account a number of requirements for measurement systems determining the quality of electricity, it is based on a simplified measurement scheme. Measurement systems should take into account not only the requirements for limits, but also for equipment contained in a number of different standards (including EN 61000-4-7, EN61000-4-30, etc.). In addition, the description of the measurement system presented in the article is given quite inconsistently, e.g.: Fig. 1 should show the architecture of the measurement system in accordance with the description, and only shows the part of the system regarding current measurement (no voltage measurement channel). Two proposed voltage measurement solutions are presented, however, one of them may pose a serious threat to the system. According to the reviewer, the direct, non-isolated measurement of voltages in switched-mode power supply environments, proposed in one of the solutions, may be a serious threat to the entire system. Errors were made in the selection of current sensors because, according to the technical data, YHDC SCT 013-10 has only a bandwidth of up to 1 kHz and a minimum bandwidth of up to 2 kHz is required, while the second proposed element of Talema AZ-500 has a bandwidth of only 50 Hz. Also in terms of verifying the system operation, the authors suggest that they checked the correctness of the system up to 3 kW, as shown in Fig. 16, however, the description shows that only in the range up to 1 kW the system was verified (the CVM-430 D32 system meets the requirements only in the range up to 1 kW and no description of how the system was verified in the band from 1 kW - 3 kW?). According to the reviewer, figures 14, 15, 16 should be made in other scales (maybe logarithmic?). In addition, the authors should change the title because it suggests that the system can be used as a replacement for energy measurement systems, and the measurement system presented in the article, unfortunately, does not meet a number of requirements for such systems and in its current form is not suitable for printing.

Reviewer 3 Report

The proposed power metering infrastructure for real time analysis is very interesting for the hardware engineers and the scientists in energy field.

This paper should be added with different approaches to the power metering. It will be better to compare different existing metering systems and define its advantages and disadvantages.

And also, you have to decide this article is about metering equipment or is about obtained result analysis? It’s a good and interesting technical paper but not a scientific article.

Round 2

Reviewer 2 Report

The current version is OK